# Influence of Nanocellulose Structure on Paper Reinforcement

**DOI:** 10.3390/molecules27154696

**Published:** 2022-07-22

**Authors:** Waldemar Perdoch, Zhuoran Cao, Patryk Florczak, Roksana Markiewicz, Marcin Jarek, Konrad Olejnik, Bartłomiej Mazela

**Affiliations:** 1Faculty of Forestry and Wood Technology, Poznań University of Life Sciences, Wojska Polskiego Str. 28, 60-637 Poznań, Poland; waldemar.perdoch@up.poznan.pl (W.P.); zhuoran.cao@up.poznan.pl (Z.C.); 2NanoBioMedical Centre, Adam Mickiewicz University, ul. Wszechnicy Piastowskiej 3, 61-614 Poznań, Poland; patryk.florczak@amu.edu.pl (P.F.); roksana.markiewicz@amu.edu.pl (R.M.); marcin.jarek@amu.edu.pl (M.J.); 3Centre of Papermaking, Lodz University of Technology, Wólczańska 221, 93-005 Lodz, Poland; konrad.olejnik@p.lodz.pl

**Keywords:** nanocellulose, paper conservation, tissue reinforcement, nanocellulose modification

## Abstract

This article describes how crystalline or fibrous nanocellulose influences the mechanical properties of paper substrate. In this context, we used commercially available cellulose nanocrystals, mechanically prepared cellulose nanofibers dispersed in water or ethanol, and carboxy cellulose nanofibers. Selective reinforcement of the paper treated with the nanocellulose samples mentioned above was observed. The change in the fibre structure was assessed using scanning electron microscopy, roentgenography, and spectroscopy techniques. In addition, the effect of nanocellulose coating on physical properties was evaluated, specifically tensile index, elongation coefficient, Elmendorf tear resistance, Bendtsen surface roughness, Bendtsen air permeability, and bending strength. It can be concluded that the observed decrease in the strength properties of the paper after applying some NC compositions is due to the loss of potential disturbances in hydrogen bonds between the nanocellulose dispersed in ethanol and the paper substrate. On the other hand, significantly increased strength was observed in the case of paper reinforced with nanocellulose functionalized with carboxyl groups.

## 1. Introduction

Paper products are used in many different areas of human activity (e.g., packaging, hygiene purposes, documents, books, drawings, maps). Transferring and archiving information has always been one of the most important tasks of paper materials. Paper documents, manuscripts, and books can be kept for hundreds of years without special conservation efforts, while the life of most electronic media is only from several to several dozen years [1,2]. However, paper items are adversely affected by negligence, wilful damage, or the inevitable decay caused by the effects of past production methods (e.g., the use of aluminium salts), storage in poor conditions, and daily human use [3,4,5]. The growing awareness of the need to preserve cultural artifacts means that more and more attention is paid to extending the life-span of paper products that are intended for carrying information. This applies to both new ones and those created some time ago.

The past 50 years in particular have seen more attention paid to paper conservation. For example, cultural heritage objects made of paper are hardened and stabilized during conservation works to stop biological degradation and limit the damage caused by variable chemical or physical factors. Activities for paper conservation can be categorized into intervention, disinfestation/disinfection, cleaning, stabilization (deacidification), repairs, and consolidation [6]. The intervention involves the characterization of appropriate material and its condition, treatment planning and documentation, isolation of the artefact, and separation from other materials. For paper repairs, replacing mechanically damaged paper with Japanese paper is a typical procedure. Japanese paper is thin and strong, derived from several shrub species, and has long been a superior material for repairing paper [6,7]. Many natural and synthetic polymers can be used for paper consolidation and strengthening through lamination, impregnation, sizing, etc. Zervos and Alexopoulou [6] reported on several important synthetic polymers used in paper conservation, either as adhesives or consolidants, including polyvinyl acetate, acrylic, polyvinyl alcohol, paraloid, polyethylene, regnal, parylene, and soluble nylon. Some natural polymers have been used for paper consolidation, e.g., starch [8,9,10,11], gelatine [12,13,14], and chitosan [15,16,17,18], to improve their mechanical and barrier properties, aging resistance, bacterial resistance, colour stability, etc. 

In recent years, application of nanocellulose (NC) has become one of the most promising and widely studied solutions for paper consolidation, strengthening, and improvement of barrier properties [19,20,21,22,23,24,25]. As the main constituent of plant fibres, cellulose is renewable and the most abundant organic compound on earth. Its abundance in plants varies from 30–50% in woods to 90% in cotton [26]. The emergence of nanocellulose has given birth to a new generation of materials that have the potential to meet the demands of modern society for high-performance materials from renewable resources. NC is usually characterized by nanoscale lateral dimension (5–50 nm) made from cellulose isolated or produced from plants, tunicates, algae, bacteria, recycled paper products, and other biomass residual streams. There are two main processes used for production of NC: the bio-formation of cellulose by bacteria (bottom-up) and the disintegration of cellulosic fibres by mechanical or chemical means (top-down) [20,27,28,29,30,31,32]. NC is generally classified into three major groups: cellulose nanocrystal (CNC), cellulose nanofibril (CNF), and bacterial nanocellulose (BC) [33]. The outstanding strength of NC has caused great interest in its use as a polymer reinforcement [34,35,36,37]. Indeed, one of the most important and unique aspects of NC is its mechanical properties. For example, the tensile strength of this material can reach values in the range of 7.5–7.7 GPa, which is much higher than steel wire (4.1 GPa) and Kevlar (3.5 GPa). NC has a high aspect ratio (L/d, where L is the length and d diameter) of up to 100, with low density (1.6 g cm^−1^) and potentially high chemical activity. Even though some inorganic nanoparticles (e.g., graphite) are characterized by the similar properties, NC has other attractive features that inorganic nanoparticles do not (e.g., better reversibility, biodegradability, no toxicity) [26,27]. BC has been recommended for human medical applications. In general, it has excellent potential as a biomedical engineering material [38,39]. NC barriers’ highly controllable rheological properties open numerous possibilities for applications such as food packaging, paint additives, and stabilizing cosmetics [40,41,42,43].

Considerable interest in NC has also been drawn in paper conservation. In a study by Santos et al. [44], the mechanical properties of paper in a book treated with BC were as good as those obtained with Japanese paper. Such treatment, however, resulted in marked improvement of optical properties compared to those restored with Japanese paper. As a result, letters on pages treated with BC were more legible than those with Japanese paper. In addition, paper restored with BC had decreased porosity. A similar study by Dreyfuss-Deseigne [45] compared NC film to different types of Japanese paper. NC film provided excellent stability to light, temperature, and aging. The unique transparency properties of NC film do not change with aging. NC film is skinny but, at the same time, stronger than Japanese paper. Thanks to previous research, NC combined with starch and chitosan has recently been employed to consolidate paper [46,47,48].

Moreover, CNC and CNF were compared by evaluating tensile strength properties, the most common properties measured in paper conservation studies, even though not related to paper usability. As a result, tensile strength is not typically used as a criterion for evaluating the suitability of paper conservation methods. In addition, NC has provided better consolidation for canvas [49,50]. The performance of pure CNC and CNF dispersions used as paper coatings has been evaluated using mechanical and barrier properties. Mechanical properties evaluated in past studies include tensile strength and folding endurance; the latter is a better expression of the usability of paper than other mechanical properties [7]. The use of nanocellulose as a strengthening agent in paper is due to the fact that it is the same chemical compound that is responsible for the formation of bonds in the paper structure.

The mechanical properties of paper depend on the degree of mutual bonding of fibres in the material structure. The strength of mutual bonding depends on the size of the bonded area and the strength of individual bonds. The main forces determining the strength of interactions between the fibres in the paper structure are hydrogen bonds, van der Waals bonds, and frictional forces [51,52,53,54].

Hydrogen bonds are often suggested as playing the most significant role in creating a solid paper structure. This bonding occurs primarily between hydroxyl groups of cellulose fibres. One cellulose molecule’s positively charged –OH particle is attracted by polarization to another molecule’s negatively charged oxygen atom. These relatively strong intermolecular interactions arise when the distance between two hydroxyl groups is about 0.26 nm. Therefore, introducing cellulose particles with nanometric size should contribute to filling the voids in the fibre–fibre contact area, which will increase the total number of hydrogen bonds formed. Consequently, an increase in paper strength should be obtained.

The aim of this work was to investigate the influence of microcrystalline cellulose and nanocellulose on the strength properties of the treated paper substrate (i.e., Whatman blotting paper). The scope of the work, unlike many previous ones, included two types of media (i.e., water and ethanol) in which microcrystalline cellulose was dispersed. It was assumed that this would have a significant impact on the tested properties. Additionally, the fibrous nanocellulose was functionalized with carboxyl groups in order to be able to assess the degree of penetration of the nanocellulose preparation into the structure of the paper substrate. In this way, it was intended to assess the ability of the substrate to adsorb nanocellulose. 

## 2. Results and Discussion

### 2.1. Nanocellulose Structure

Four types of cellulose nanostructures were evaluated for their influence as paper reinforcement additives: cellulose nanocrystals (CNC), cellulose nanofibers prepared in water (CNF(W)), or ethanol (CNF(Et)), and cellulose nanofibers functionalized with carboxyl groups (CNF-COOH). The structural properties of each nanocellulose additive were examined using SEM micrographs (Figure 1, Figure 2, Figure 3 and Figure 4).

The structure of CNC, shown in Figure 1, did not exhibit the expected presence of nanocrystals. Instead, the sample was fibrous rather than possessing rod-like or whisker shapes, with a broad range of fibre diameters. CNC consisted of a multi-layered fibre network instead of primarily individual nanocellulose fibres. The smallest fibres were from 30 to 120 nm in diameter (Figure 1b). As CNCs are obtained through chemical treatment, their amorphous regions have been lost, and as a result, their structure is composed of crystalline fibre regions only [47].

Figure 2 shows CNF(W) obtained via ultrasonic treatment. Ultrasonic treatment directly disrupts hydrogen bonds of cellulose fibres by mechanical means, allowing the preparation of smaller particles. Moreover, fibres of mechanical origin are composed of both amorphous and crystalline cellulose regions, contrasting with chemical treatment, in which amorphous regions are lost [55]. In the CNF(W) case, it was possible to obtain fibres more homogenous in size and shape, with diameters from 25 to 65 nm, as shown in Figure 2b. Importantly, CNF(W) nanocellulose forms single fibres, which was not the case for commercially purchased CNC.

CNF(Et) samples were prepared similarly to CNF(W) samples however, due to different dispersion environments, they differed in properties. In this case, CNF(Et) fibres were more uniform (Figure 3). However, the sample drying process resulted in a more agglomerated final product, which may be caused by faster solvent removal, allowing submolecular interactions between cellulose hydroxyl groups to occur, yielding a fibrous film. In this case, the self-assembly of the CNF(Et) structures may be promoted by ethanol, as this solvent is used as a coagulator of nanocellulose material [56]. The diameters of the visible fibrils ranged from 30 to 100 nm.

As shown in Figure 4, CNF-COOH also exhibited a fibrous structure, with minimal diameters, reaching from 12 to 28 nm, and was thinner, with long rod-like shaped structures. Importantly, obtaining this material through acid hydrolysis allows, as previously mentioned, the removal of amorphous regions.

CNF(W) and CNF-COOH(W) are presented in Figure 5, and both exhibit the main characteristic bands in cellulose units. The absorption band at around 3300 cm^−1^ comes from –OH stretching vibrations. 2900 cm^−1^ and 1370 cm^−1^ bands can be assigned to stretching and deformation vibrations of C–H groups in glucose units. C–O–C and C–O stretching vibrations are visible in the region of 1000–1200 cm^−1^. The region of 1200–1500 cm^−1^ is where the structural transformation of cellulose chains might be revealed. Moreover, the small absorption band at ca. 900 cm^−1^ is characteristic of glucose unit β-glycosidic linkages. Notably, one can see that the CNF-COOH spectra revealed a peak at 1726 cm^−1^, confirming cellulose fibre oxidation and the presence of carboxyl groups [56,57].

The X-ray diffraction patterns of microcrystalline cellulose, nanocellulose fibre (CNF(W)) and nanocellulose functionalized with carboxyl groups are presented in Figure 6. A very well-defined main peak around 20–25° (2Theta) in all patterns is visible, which means that the main form of cellulose is cellulose type I [58]. In each case, the peaks are sharp and narrow, which proves the high crystallinity of all samples.

The XRD data were used to establish the crystallinity indices (CrI) of the prepared nanocellulose materials together with the microcrystalline cellulose, using the deconvolution method. The crystalline peaks located around 16° and 22° were separated from the amorphous region around 18.5°, and the crystallinity percentage was calculated from the ratio of the integrated area of crystalline peaks to the total integrated area under the XRD signals. In this way, the crystallinity indices for the microcrystalline cellulose, nanocellulose fibres (CNF(W)) and nanocellulose fibres functionalized with carboxyl groups (CNF-COOH) were calculated to be equal to 88.8%, 83.8% and 94.1%, respectively. Slightly lower crystallinity for the CNF(W) sample was related to mechanical treatment breaking the crystallinity; on the other hand, the higher crystallinity of the CNF-COOH confirmed the breaking of the remaining amorphous region with the acid treatment. Such behaviour is consistent with the literature data [59].

### 2.2. Structural Properties of Treated Paper

Figure 7 shows the apparent density of WBP, from which it is possible to deduce an increasing tendency after nanocellulose treatment. The apparent density of most samples was similar, but the values for CNC- and CNF-coated papers were higher than the other variants by about 0.015 g/cm^3^. Among papers treated with a nanocellulose coating, the lowest apparent density was obtained for paper coated with CNF(Et) (Figure 7), which is attributed to the low polarity of the ethanol solvent. Ethanol’s dipole moment is lower than that of water, reducing the energy of hydrogen bonds and lowering the adhesion of CNF to paper. This follows the theory that more polar solvents create stronger hydrogen bonds. The highest NC retention, expressed by the highest observed apparent density, was found in CNC(W), i.e., fibrous cellulose with the greatest variation in nanoparticle size, as visualized by SEM. Strong adhesion of nanocellulose to paper occurred most likely thanks to particle sizes being better matched to the surface and structure of the base material (WBP). 

The CNC(W)-coated paper had a higher roughness than the CNF-coated papers and uncoated paper (Figure 8a). In addition, the high retention of CNC on the paper surface may have influenced the increase in roughness. This phenomenon most likely resulted from the CNC spatial structure and non-uniform particle size (Figure 8a). Nanocellulose suspended in ethanol had slightly less air permeating the paper structure. Despite the low retention of CNF(Et), paper coated with this NC was characterized by high roughness, which could be due to weaker hydrogen interactions compared to water-containing systems. The surface roughness of untreated paper differed between side A and side B due to the production process of WBP sheets (under-screen and over-screen sides). Side A and side B maintained the same trend for CNC(W) and CNF(Et), with side A smoother than side B, but the surface roughness of CNF(W) and CNF-COOH was similar for both sides of the paper.

The greatest decrease in air permeability was observed for CNC-treated paper (W). This sample showed about 90% lower permeability compared to the non-treated paper. This phenomenon can be explained by the effect of filling the porous structure of the paper with a large amount of CNC (W) particles. The low permeability of CNC-modified paper (W) is also attributed to the much more considerable amount of NC on the paper’s surface (Figure 7). According to Zimmermann et al. [57], hydroxyl groups exposed to the nanocellulose surface favour the formation of nanofiber networks, thanks to the formed hydrogen bonds. The number of hydrogen bonds increases due to the larger contact surface between the nanoparticles and the fibres. The new bonds increase the apparent density of the material, making it more resistant to air and gaining mechanical properties. This theory confirms the relationships observed in our study and the differences between the CNC and other variants [55,56,58]. Substitution of the COOH group in CNF does not change the strength properties of the treated paper. According to the above theory, CNF-COOH contains a similar number of exposed active groups as the paper treated with CNF (W).

### 2.3. Mechanical Properties of Paper

Figure 9a,b present a comparison of the paper tear resistance index as a function of the tensile index.

Tensile strength and tear resistance were tested as paper materials’ most important mechanical parameters. The most common improvement in raw materials for papermaking used in the industry—based on the refining operation—results in an increase in the tensile strength and a simultaneous decrease in the tear resistance. For this reason, both of these parameters are often presented together. Measurements were performed for MD (machine direction) and CD (cross-machine direction). Except for WBP coated with CNF-(Et), the tensile index in MD was about twice that of CD (Figure 9a,b). Application of CNF(Et) to the paper surface significantly reduced the tensile index and, at the same time, increased tear resistance (from about 87 mN to about 113 mN for MD and from about 101 mN to 120 mN for CD). The lower dipole moment of ethanol relative to water was the likely cause of reduced paper breaking strength, which was ca. 40% lower for CNF(Et) with a tensile index of 17.68 m×N/g compared to 29.54 m×N/g in the untreated paper. According to the literature [39], the reduction in dipole moment is attributable to weaker hydrogen bonds. The magnitude of this energy depends on the solvation conditions and the geometric requirements for the formation of hydrogen bonds. In this theory, higher polarity solvents, such as water compared to ethanol, form stronger hydrogen bonds [40,41,42]. The greatest mechanical strength in both directions (MD and CD) was a characteristic of paper treated with CNC(W). In that case, the tensile index increased to ca. 36 m×N/g for MD and almost 18 m×N/g for CD. At the same time, the tear resistance for MD increased to 102.4 mN and for CD to 136 mN. Presented results show that not every composition containing nanocellulose will increase the strength properties of the paper. Similar conclusions can be drawn from the results presented in Figure 10 and Figure 11.

The elongation of the CNC(W)-coated paper was increased by 25% in MD, as shown by the tensile index (Figure 10). This can be attributed to the structure of the nanocellulose introduced, i.e., CNC(W) has a much broader spectrum of nanocellulose particle sizes (30–120 nm), as shown by SEM analysis, which resulted in higher retention on the base paper.

Although no significant change in tensile strength was observed after protecting WBP with CNF(W), the MD and CD elongation of papers with CNF(W) and CNF-COOH coatings was higher compared to the reference sample (WBP). In the case of CNF-COOH, this can be explained by additional carboxyl groups, which positively affected this aspect of paper strength.

Nevertheless, it is noted that the highest efficiency in improving paper strength properties was demonstrated by paper treated with CNC(W). This is shown by tensile index and tear resistance in Figure 9a,b. The highest average breaking force was needed to damage samples with the CD’s highest tensile index. The force required to tear the CNC(W)-coated paper was about 25% higher than that required to tear the non-coated paper.

Interestingly, paper samples coated with CNF(W) and CNF-COOH(W) showed lower tensile and tear resistance index values despite their high elongation.

Tear resistance index depends mainly on the length and strength of the cellulosic fibres and, to some extent, the strength of bonds between the fibres. It was observed that the treatment with CNC(W) and CNF(Et) increased paper tear resistance index in both MD and CD directions compared to non-coated paper.

There was no significant difference between untreated paper and CNF (W) or CNF-COOH treated paper. However, it is puzzling why these differences are blurred even in the case of CDs. Presumably, CNF, as an additional layer of non-oriented fibres, in this case, acted as an additional reinforcement that covered the weak side of the CD direction (Figure 9a,b).

Figure 11 shows the averaged stress-strain curves for all tested samples. Figure 11a contains the results of measurements performed in the MD direction, and Figure 11b in the CD direction. The analysis of these curves allows one to confirm that the application of the CNC(W) composition resulted in the greatest increase in both the elongation and tensile strength of the paper. Increasing these parameters increases the ability of the material to absorb energy during its stretching. The improvement in elongation, but without a significant increase in tensile strength, was also visible for the CNF(W) and CNF-COOH compositions and in both material directions (MD and CD). The effect of the CNF(Et) application was minor.

None of the treated paper was resistant to bending forces. Regardless of the nanocellulose used and the web direction, the double-fold number varied from 4 to 8 fold (Figure 12).

Table 1 presents the coefficients of variation for all measurements conducted in the presented research.

## 3. Materials and Methods

### 3.1. Materials

Microcrystalline cellulose (MilliporeSigma, Burlington, MA, United States, Cat. No. 435236), nitric acid (Avantor, Gliwice, Poland, 65%), and sodium nitrite (MilliporeSigma, Burlington, MA, United States, Cat. No. 237213) were used without further purification. Milli-Q^®^ (MilliporeSigma, Burlington, MA, United States) deionized water with the resistivity of 18 MΩ cm^−1^ was used for each synthetic and purification step. Whatman^®^ quantitative blotting papers (WBP), grade 40, with a density of 0.468 ± 0.01 g/cm^3^, were used as a base cellulose material. An aqueous suspension of cellulose nanocrystals CNC(W) (7.4 wt%) was purchased from BGB in Canada. Ethanol-based cellulose nanofibers CNF(Et) (3%), water-based cellulose nanofibers CNF(W) (3%), and water-based fibrous nanocellulose, functionalized with carboxyl groups CNF-COOH (2%) were prepared according to the following protocols:

#### 3.1.1. CNF(W) and CNF(Et) Preparation

Ultrasonic treatment was performed using an ultrasonic generator (BRANSON 550 MilliporeSigma, Burlington, MA, United States). Water (CNF(W)) and ethanol (CNF(Et)) dispersion of microcrystalline cellulose (3 wt. %) without any chemical additions were prepared with the use of high-intensity ultrasonication at 550 W for 5 h in an ice-water bath with a distance from the tip of the ultrasonic generator probe to beaker bottom of about 20 mm.

#### 3.1.2. CNF-COOH(W) Preparation

Fibrous nanocellulose, functionalized with carboxyl groups, was obtained according to the previously described protocol [57], with slight modifications. In this procedure, 56 cm^3^ of nitric acid (65%) and 10 g of microcrystalline cellulose were placed in a round bottom flask, after which 0.96 g of sodium nitrite was added. The reaction mixture was then thoroughly mixed for 12 h at 50 °C. Subsequently, 250 cm^3^ of deionized water was added to the reaction mixture to quench the product reaction. The first portion of supernatant liquid was removed. The product was further washed with the ethanol–water mixture (ratio 1:2) and separated from the solid product until the supernatant pH was approximately 5. The fibres obtained were sonicated in water (CNF-COOH concentration of 2%) for 2.5 h to obtain a good dispersion of nanofibers.

### 3.2. Nanofiber Structure and Morphology 

Scanning electron microscope (SEM) images were collected using a Jeol 7001TTLS scanning electron microscope (Jeol Ltd., Tokio, Japan). Attenuated total reflectance Fourier transform infrared spectroscopy (ATR-FT-IR) spectra were collected from 4000 to 400 cm^−1^ at a resolution of 1 cm^−1^ using a Jasco FT/IR-4700 spectrometer (Jasco Corporation, Tokio, Japan). FTIR spectra were collected for bulk cellulose material. Powder X-ray diffraction (XRD) studies of the cellulose material was carried out on an Empyrean (PANalytical, Malvern, UK) diffractometer, The XRD diffractometer uses Cu Kα radiation (λ = 1.54 Å), a reflection-transmission spinner (sample stage), and a PIXcel 3D detector, operating in the Bragg–Brentano geometry. Scans were recorded in angles ranging from 20 to 90° (2Theta) with a step size of 0.006 and continuous scan mode, at room temperature.

### 3.3. Sample Treatment 

Dispersions of CNC, CNF(W), and CNF-COOH(W) were diluted with water to a final concentration of nanocellulose equal to 0.15%. Similarly, CNF(Et) was diluted with ethanol to reach the same concentration. Dispersed NC was homogenized using a Unidrive X 1000 (CAT Scientific, Tübingen, Germany) equipped with a T6 shaft at around 15,000 rpm. Samples of WBP were treated with the abovementioned dispersions using the soaking method (5s). Samples were dried in constant conditions (25 °C, RH 50%) for 14 days. Sample dimensions measured depended on the type of test.

### 3.4. Paper Testing

All paper samples were conditioned according to ISO 187:1999. Properties were determined in accordance with relevant ISO standards:– Tensile index and elongation (ISO 1924-2:2008),– Elmendorf tear resistance (ISO 1974:2012), – Bendtsen surface roughness (ISO 8791-2:2013),– Bendtsen air permeance (ISO 5636-3:2013), – Folding endurance (Schopper device—ISO 5626:1993), and– Apparent density (ISO 534:2005). 

Tensile index and elongation tests were carried out using an INSTRON tensile testing machine, model 5564 (High Wycombe, UK). Tear resistance was determined with the Elmendorf tester (Testing Machines Inc., New Castle, DE, US). All strength properties were performed in both paper directions (Machine Direction (MD) and Cross Direction (CD)). Surface roughness and air permeance of the samples were measured using a Bendtsen tester made by TMI Group (Buchel B.V., New Castle, DE, US). Folding endurance (Double Folds Number) tests were performed using a Schopper type folding tester made by Labor-Meks, Łódź, Poland. Apparent density was determined using a Lorentzen & Wettre, type 222 digital micrometre, with 10 samples of each paper treatment cut into 10 × 10 cm pieces and weighed. Sample thickness was measured with an accuracy of 1 µm at 10 random locations on each paper sample.

The method and materials used in the paper methodology are graphically presented in Figure 1.

## 4. Conclusions

It was found that the structure of nanocellulose preparations affected paper strength. The apparent density of paper treated with nanocellulose proved to be the most reliable indicator of how well a nanocellulose formulation penetrated the paper structure. The amount of nanocellulose introduced into the paper was determined by the presence or absence of different functional groups and by different sizes of nanoparticles. This strongly affected tensile index, tear resistance, and elongation properties. The medium (water or ethanol) in which cellulose nanoparticles were suspended was equally important. Probably the use of a solvent with lower polarity (e.g., ethanol) caused a reduction in the strength of hydrogen bonds, thus reducing all strength parameters of the modified material.

Based on the results shown in Figure 9 and Figure 10, it cannot be concluded that CNF-COOH gave better elongation and higher tensile index results than the other variants and the control. Due to insufficient retention of the NC samples, no significant increase in strength parameters resulting from the modification of cellulose nanoparticles by implementing additional carboxyl groups was shown. The obtained results showed that the presence of nanocellulose improved the strength properties of the paper but not in all cases.

## Data Availability

The data presented in this study are available on request from the corresponding author.

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
