# Peer review of "Influence of Nanocellulose Structure on Paper Reinforcement"

_molecules, 2022, doi:10.3390/molecules27154696_

Round 1

Reviewer 1 Report

Dear authors, I congratulate you for the excellent work. My observations are unique and may contribute to better points. I think that (i) a quantitative methodology should be added for the -COOH groups to better assess the influence of these groups on the observed properties. (ii) Since FTIR is dependent on the concentration of the sample, it would be interesting to make divisions between the intensities of bands within the same spectrum, which would allow later to compare the variations between the different spectra. (iii) I think that determine the crystallinity indices of the samples and only then, if the values ​​are different, try to correlate with the mechanical properties. What would avoid keeping the discussion only in function of the polarity of the solvents. (iv) in item 3.2. Structural properties of treated paper - I particularly understand that in the different graphs, the dispersion bars for the different samples coincide within an interval (smallest and largest), which makes me understand that there is no differentiation in the responses, however, this differentiation only is perceptible by the average values. However, I think that giving a statistical treatment with a confidence interval of 95% or 90%, could really show which groups are distinct from the others and thus, improve the discussions.

I say goodbye by congratulating everyone.

Author Response

We thank the reviewer for every comment, which made us revise the article and improve its quality. Point-by-point responses to all the comments can be found below. All changes are moreover highlighted within the revised version of the manuscript.

Reviewer 2 Report

The manuscript by Mazela and co-workers concerns the reinforcement of paper by nanocellulose. The manuscript reads well and could be published in this journal, but the interest and the originality are not very high. However, while preparing the revised version of the manuscript the authors should take into account the following, minor, suggestions:

-line 39: please cite some basic reference on the subject instead of, or together with, reference 3.

-line 46-48: please introduce some basic reference on the subject

-line 122-125: after a long introduction, the authors dedicate only one sentence to the aim of the manuscript. Please add something more trying to “sell” you research, as an example, reporting what is supposed to be your improvement with respect to the state of the art. Is the first time this approach has been proposed? If no, what are the differences? Etc.

-page 5-6, figures 1-4: are the SEM images representatives? Moreover, try to improve the contrast and the general quality of the images

-line 244: change the term spike with peak (or any other term used in spectroscopy)

-figure 7: how did the authors measure the density? What is the precision of the method? Maybe the differences between samples are not meaningful

Author Response

(The authors gave the same response as above.)

Reviewer 3 Report

In this work, paper substrate was treated by crystalline or fibrous nanocellulose, and the mechanical properties were evaluated. Despite some valuable results, there are lots of problems that should be clarified and more experiments and discussion should be supplied before further consideration.

1. In experimental section, the authors claimed that samples were dried in constant conditions (25°C, RH 50%) for 14 days. The processing time is a bit long. It seems to be inefficient. Why didn’t the authors use an oven?

2. Please pay attention to Figure 7-11. There are no scale lines.

3. The resolutions of Figure 1-5 are not high. Please improve their quality.

4. Please supplement the characterizations of treated papers including SEM and FTIR to confirm the successful modification.

5. Can the authors evaluate the durability and water resistance of the treated papers?

6. Can the authors supply the stress-strain curves for these samples?

Author Response

(The authors gave the same response as above.)

Round 2

Reviewer 3 Report

This manuscript has been improved by the authors. It can be accepted for publication.